# Discovering the Landscape and Evolution of Responsible Research and Innovation (RRI): Science Mapping Based on Bibliometric Analysis

**Jiqing Liu [1], Gui Zhang [2,*] , Xiaojing Lv [1] and Jiayu Li [3,*]**

[1]  School of Economics and Management, Hebei University of Technology, Tianjin 300401, China;
      skylar_liujq@126.com (J.L.); lv_xiaojing@yeah.net (X.L.)
[2]  College of Beijing-Tianjin-Hebei Collaborative Development, Nankai University, Tianjin 300071, China
[3]  School of Public Policy & Management, Tsinghua University, Beijing 100084, China
[*]  Correspondence: zhanggui2050@126.com (G.Z.); ljyu@tsinghua.edu.cn (J.L.)

**Abstract:** The growing number of papers on Responsible Innovation (RI) and Responsible Research and Innovation (RRI) have shaped the popularity and usefulness of RI and RRI as a technology governance concept. This study reviews and assesses the development of RRI research through a bibliometric analysis of 702 RRI-focused papers and 26,471 secondary references published in the Web of Science Core Collection database between 2006 and 2020. Firstly, the paper provides a broad outline of the field based on annual growth trends, journal distribution, and disciplinary distribution for RRI publications. Secondly, this study reveals the current state of RRI research by identifying influential literature, journals, authors, countries, and institutions. Thirdly, a phased keyword analysis is conducted to determine the stage characteristics of the RRI field. Finally, based on the bibliometric analyses, this study summarises the evolutionary trajectory of RRI and makes recommendations for future research directions. As a complement to the previous qualitative literature review, the paper provides a systematic and dynamic understanding of RRI research.

**Keywords:** responsible innovation (RI); responsible research and innovation (RRI); bibliometric review; CiteSpace; stakeholder engagement; ethics; public engagement; technology assessment; sustainability; science policy

## 1. Introduction

Responsible Innovation (RI) and Responsible Research and Innovation (RRI) are important concepts that have been developed rapidly in recent years and are receiving increasing attention. RI and RRI advocate the integration of ethical values with 'responsibility' as a core dimension of research and innovation (R&I) policy and practice so that innovation can meet both social expectations and ethical standards [1], and achieve a sustainable future through socially desirable methods [2]. The conceptual definitions, tools, and methods of RI and RRI have been greatly expanded by researchers, including scholars, policymakers, and experts in specific technical fields. With Horizon Europe taking over from Horizon 2020 as the R&I funding programme of the European Union (EU), RRI is moving from the mainstreaming stage to becoming embedded in practice and, therefore, entering a new chapter of development.

The RI and RRI concepts were born out of a deep social context. An accelerated economic pace and greater industrialisation have led to a growing problem of resource scarcity and environmental damage. Moreover, some emerging technologies, associated with ethical and risk issues, have faced significant public controversies, such as genome editing [3–5], artificial intelligence [6,7], and nuclear energy [8–10]. The traditional separation between technology developers and those impacted by technology has also been challenged, in terms of both power asymmetries and temporal asynchronies. The 'Collingridge dilemma' clearly

summarises the technological governance complexity, namely, that the social consequences of technology cannot be anticipated at an early stage, yet, by the time the undesired social consequences are discovered, the technology has often become part of the overall economic and social fabric, making it difficult and expensive to control [11]. As a result, the relationship between science and society is being re-examined and redefined. Science and society are becoming less independent and more integrated and symbiotic. Correspondingly, both the hard sciences and the social sciences and humanities also require more interaction and exchange of knowledge across boundaries [12,13]. Specifically, R&I that rely solely on market regulation and self-monitoring are considered 'weak' when considering their social benefits, a factor that explains the trend towards social embedding of innovation, known as socio-technical integration. Socio-technical integration is ultimately linked to RRI [14], as it seeks to incorporate social research into the R&I process in emerging, or potentially controversial, technological areas.

After more than a decade of development, the field of RRI has become more mature. Notably, several review articles summarising key issues in its development have been published. Burget, et al. [15] conducted a systematic review of the definition and concept of RRI based on 235 papers covering six dimensions: inclusion, anticipation, responsiveness, reflexivity, sustainability, and care. Schuijff, et al. [16] performed a systematic review of RRI practices based on 52 papers, covering its main methodological and value dimensions. Thapa, et al. [17] provided a focused review of the application of the four domains of RRI—drivers, tools, outcomes, and barriers—to regional innovation and regional sustainable development, while Harsanto, et al. [18] offered a preliminary review of the current state of research on RRI in emerging economies. In addition to these qualitative reviews, Timmermans [19] employed a quantitative approach to map the RRI landscape, though the dataset only covered the years up to 2014. While collectively these reviews certainly improve the understanding of RRI, each focused on different aspects of the definition, methodology, application, and new contexts of RRI and so there is a lack of overall comprehensiveness. Moreover, the qualitative reviews have limitations, such as small sample sizes, subjectivity, and lack of rigor [20].

Given the narrative nature and time constraints of the extant reviews, a more extensive quantitative review remains necessary. In this study, bibliometric analysis methods were used to provide a more comprehensive review of RRI, thereby complementing the qualitative reviews in several ways. Firstly, bibliometric analysis is widely used to analyse the publication performance of individuals and institutions [21]. To assess the performance of RRI research, the first research question (RQ1) is: What is the current state of RRI research? Secondly, bibliometric analysis methods can calculate the knowledge structure of the knowledge space [22], which leads to the second research question (RQ2): What are the hotspots of RRI research? Based on RQ1 and RQ2, the third research question (RQ3) is posed: What are the future directions of RRI practice?

Based on a sample of 702 publications, bibliometric research methods were used to perform a systematic and comprehensive quantitative review of RRI and, thereby, contribute to research in the RRI field. Firstly, the landscape of this field was mapped by analysing the following: (1) annual trends, journal distribution, and disciplinary distribution of publications; (2) several co-citation analyses of authors, references, and journals, respectively; and (3) the institution co-authorship network and the country co-authorship network. This analysis revealed the current status, hotspots, and knowledge structure of the extant body of RRI research. Secondly, the development of RRI longitudinally—that is, along the time dimension—was analysed. Based on keywords analysis, this study divided the development process of RRI into three phases and summarised the research themes and research characteristics of each phase. This approach elucidated the phase characteristics of RRI research. Thirdly, this study summarised the evolutionary trajectory of RRI and made predictions about future research directions. By synthesising the results of each analysis, the evolutionary logic of RRI was illustrated in four dimensions—theory, practice, subject,

and context—and potential themes for future research were identified. This provides an important reference for future RRI researchers.

## 2. Theory Background

RI and RRI are gaining popularity as important topics in the field of ethical governance of technology, but their constituent elements and tools were well documented even before these concepts were born. Such issues as science and technology studies (STS) [23,24]; anticipatory governance [25]; ethical, legal, and social implications (ELSI) or aspect (ELSA) studies [26]; technology assessment (TA) [27,28] and its various forms, such as constructive technology assessment (CTA), are among the more prominent and immediate academic foundations of RI and RRI. The aims of these early concepts are partly reiterated and further extended in the RI and RRI frameworks.

Although similar in core connotations, RI and RRI have different origins. To be specific, RI has strong academic roots [29,30]. The most far-reaching definition of RI was proposed by Stilgoe, et al. [2], who said it was about 'taking care of the future through collective stewardship of science and innovation in the present'—. This framework asserts that the responsibility for innovation is future-oriented, and innovation is sustainable. In addition, RI integrates specific approaches embedded in the innovation process, such as anticipation, reflexivity, inclusive deliberation, responsiveness, and openness, while focusing on ways to deal with uncertainty [2,31]. Although it has not achieved widespread acceptance, RI has been introduced into research institutions as a policy framework, most notably in the UK Engineering and Physical Sciences Research Council (EPSRC) [32].

In contrast to RI, RRI has deep roots in policy and is an accepted public policy discourse within the European Commission (EC), which has been introduced into the research field in a top–down manner. RRI is supported by EU policies, such as the 7th Framework Programme (FP7) and Horizon 2020 [33]. Among several influential definitions of RRI, the definition proposed by von Schomberg [34] covers more comprehensive elements, such as anticipation, reflexivity, inclusion, precaution, and responsiveness, in its conceptualisation, while highlighting that the early framework of RRI shares obvious semantic similarities with RI. Further, the EU has identified the thematic elements of RRI as public engagement, gender equality, science education, open access, ethics, governance, sustainability, and social justice. In contrast to other connotative frameworks of RRI, these dimensions incorporate issues of social justice.

Due to the similar connotations, RI and RRI are examined as a holistic concept in the follow-up research, and collectively referred to as RRI. The interchangeability of the two terms has been validated by their cross-application in a number of projects [33].

## 3. Research Design

Figure 1 illustrates the data collection process for this study, which took place in two stages. The first stage entailed a search for documents, while the second stage focused on screening the documents.

In the first stage, three sub-databases (SCI-EXPANDED, SSCI, and CPCI-S) in the core set of the Web of Science were used as data sources. On the one hand, 'responsible innovation' or 'responsible research and innovation' were identified as search terms in the initial literature search. On the other hand, all the literature in Journal of Responsible Innovation (JRI) were included in the dataset, as this journal is highly related to the subject of this research. In the searches, the time span was set to 'before 2021'.

In the second stage, the documents retrieved in the first stage were screened. Duplicate and irrelevant papers were excluded. To avoid affecting the results of the subsequent bibliometric analysis, the papers without abstracts were excluded.

The two-stage data collection process identified 702 RRI-related references and generated 26,471 cited references as samples for bibliometric analysis. The earliest publication year of the papers in the selected dataset was 2006, so the follow-up analysis focused on the RRI-related literature from 2006 to 2020.

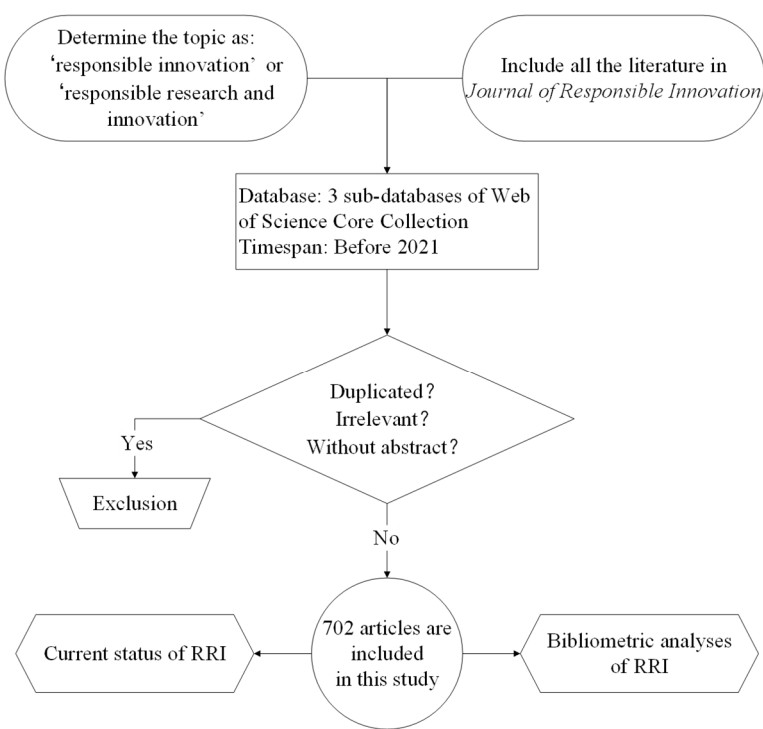

**Figure 1.** Outline of literature searching and screening process.

Bibliometrics was the main empirical method employed for this paper. Bibliometric analysis is based on scientometrics, which quantitatively analyses the key characteristics of publications in a scientific field and expresses them through scientific mapping [20,35]. This study used CiteSpace, a flexible knowledge map visualisation tool [36], to analyse the 702 collected documents based on bibliometric ideas and methods. Firstly, CiteSpace adopts a divide-and-conquer strategy in analysing the evolutionary history of knowledge domains by forming a merged network of separate co-citation networks of multiple time slices in a time series, highlighting the main changes in adjacent slices and visualising the focus in the knowledge domain [37]. Cluster analysis can also be performed [38]. Secondly, CiteSpace can detect citation bursts. Nodes with strong bursts indicate that they have attracted a lot of attention in a short period of time, which helps to find hotspots of research in a short period of time. CiteSpace is widely used for scientific mapping in a variety of fields and has proven to be a reliable tool [39–41].

This study focused on using the two main features of CiteSpace to analyse the development process and structural relationships of RRI domain knowledge and visualised the research landscape of RRI in this period through text mining, scientific analysis and mapping.

## 4. Current Status of RRI Research

### 4.1. Annual Trends in RRI-Related Publications

Figure 2 shows the distribution of RRI-related articles published in the period 2006–2020. Before 2010, research on RRI was sporadic. Although the number of papers published in 2010 was also low, the rapid increase since then has mainly been due to the launch of several RRI projects funded by the EU and other funding agencies since around 2010. The EPSRC, for example, has embedded in its funded Nanotechnologies for Environmental Solutions programme an exploration of how to incorporate approaches to promote responsible science and innovative research [42]. After 2014, the number of papers published increased significantly, indicating that RRI has been receiving more attention from the academic community. The EU's largest science and technology funding programme, known as Horizon 2020, was launched in 2014, and RRI is an important goal and topic within its

framework. Over time, the actual cases and problems arising since the implementation of the EU policy have provided a realistic basis for further research on RRI.

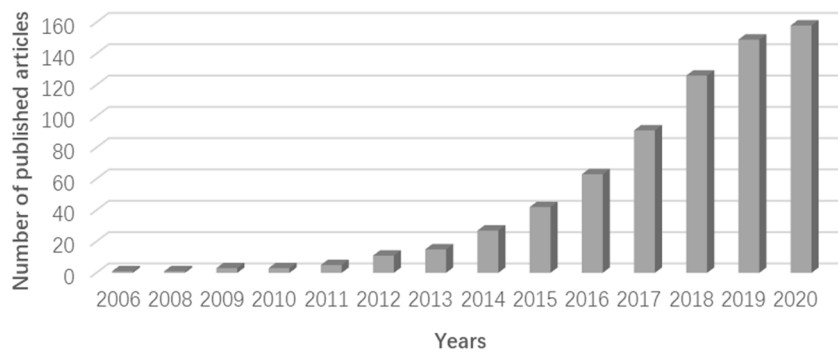

**Figure 2.** Distribution of the published articles concerning RRI over the period 2006–2020.

In addition to the ongoing RRI policies, initiatives, and projects, JRI, a journal dedicated to RRI that began publication in 2014, has driven the rapid rise in the number of articles published. JRI provides a forum for scholars and practitioners in this emerging interdisciplinary field to discuss the ethical, social, and governance issues that arise in the process of R&I, which has enabled RRI to gain broader attention and attract more far-reaching development.

### 4.2. Journal Distribution of Publications on RRI Research

Nearly 50% of the papers in the dataset were published in the top 10 journals (Figure 3). JRI is undoubtedly the journal with the most RRI-related publications, with 150 total papers, more than 25% of the dataset. This is not only because most of the papers in JRI are included in the dataset, but also because JRI focuses solely on RRI-related issues. Moreover, some pairs of papers that were not relevant to the RRI topic were excluded when building the dataset up-front, so as to ensure the validity of the results. Apart from JRI, the top 10 journals were mainly from the fields of technology ethics, public policy, and emerging science (e.g., nanotechnology, agriculture), which is in line with RRI's focus on negative externalities and ethical issues of emerging technologies at the theoretical level, as well as its focus on building broad stakeholder action networks at the practical level.

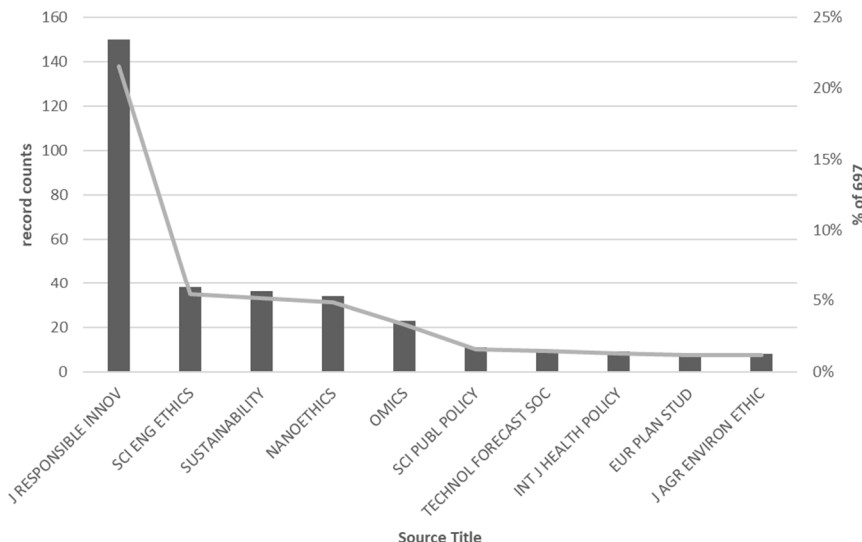

**Figure 3.** Top 10 journals with the highest number of RRI-related articles.

### 4.3. Disciplinary Distribution of RRI Research

Subject category co-occurrence analysis reveals the distribution of disciplines involved in a given domain. Figure 4 shows the results of the co-occurrence analysis of RRI subject categories after a simplification of the pathfinder algorithm. This algorithm, which is commonly used in the process of visualising knowledge structures, removes links that violate the assumption of triangular inequalities and is a compromise between the original complex network and the minimum spanning tree [43,44]. The subject category co-occurrence network of RRI research consisted of 84 nodes and 133 links. The nodes represent subject categories, and the node size is proportional to the number of studies on that subject. The nodes representing subject categories are depicted as tree rings, which provide a clear picture of the research history of the subject. The colour of the tree rings indicates when the results of the corresponding subject were published, and the thickness of the tree rings is proportional to the number of research results within the corresponding time division.

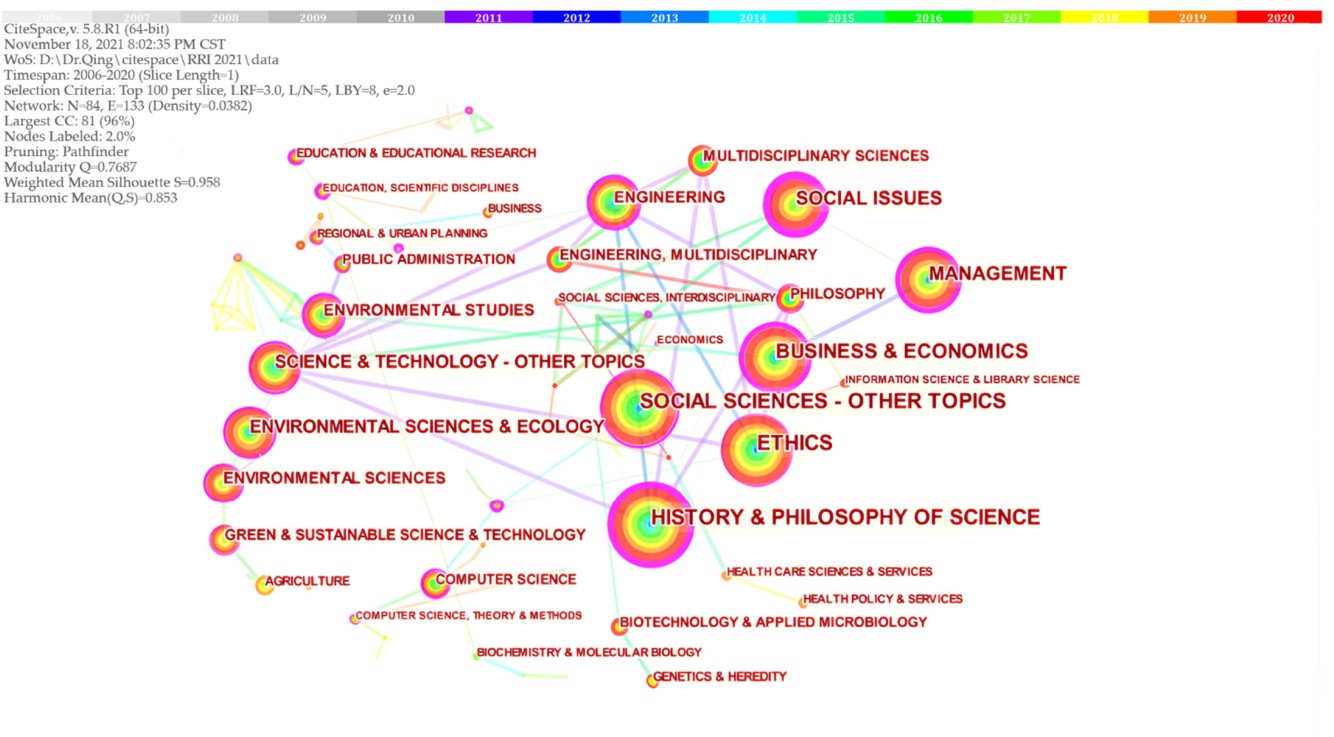

**Figure 4.** Subject category co-occurrence network of RRI research from 2006 to 2020.

The results in Figure 4 show that the categories with the highest frequencies (more than 150 papers) were Social Sciences, encompassing Other Topics (268), Ethics (252), History and Philosophy of Science (246), Business and Economics (214), Management (187), and Social Issues (164). Other important subjects with frequencies less than 150, but greater than 50, included Environmental Science—Engineering, which illustrates the interdisciplinary nature of RRI. RRI focuses on ethical issues and social governance issues of science and technology. In practice, there is a growing interest in how companies can apply RRI concepts to innovation management.

A burst analysis on the subject category was also conducted; the results are shown in Figure 5. Perhaps not surprisingly, the subject categories with strong citation bursts included a variety of technical fields, such as Engineering, Biochemistry, Computer Science, and Agriculture. Thus, RRI research appeared to be closely related to the development of specific technologies. The research field would, in turn, change as technology development evolves in a certain period.

## Top 12 Subject Categories with the Strongest Citation Bursts

| Subject Categories | Year | Strength | Begin | End | 2006 – 2020 |
|---|---|---|---|---|---|
| ENGINEERING | 2006 | 7.67 | **2009** | 2016 | |
| MULTIDISCIPLINARY SCIENCES | 2006 | 6.43 | **2011** | 2016 | |
| ENGINEERING, MULTIDISCIPLINARY | 2006 | 5.52 | **2011** | 2016 | |
| PHILOSOPHY | 2006 | 3.79 | **2011** | 2016 | |
| PUBLIC ADMINISTRATION | 2006 | 3.48 | **2012** | 2015 | |
| BIOCHEMISTRY & MOLECULAR BIOLOGY | 2006 | 4.06 | **2013** | 2016 | |
| COMPUTER SCIENCE | 2006 | 5.28 | **2014** | 2016 | |
| COMPUTER SCIENCE, INFORMATION SYSTEMS | 2006 | 2.89 | **2014** | 2016 | |
| CELL BIOLOGY | 2006 | 2.87 | **2014** | 2015 | |
| COMPUTER SCIENCE, THEORY & METHODS | 2006 | 3.47 | **2016** | 2017 | |
| EDUCATION, SCIENTIFIC DISCIPLINES | 2006 | 3.47 | **2016** | 2017 | |
| AGRICULTURE | 2006 | 3.03 | **2018** | 2020 | |

**Figure 5.** Top 12 subject categories with the strongest citation bursts over the period between 2006 and 2020.

## 5. Co-Citation Analyses on RRI Research

### 5.1. Author Co-Citation Analysis

Author co-citation analysis reveals the relationships between authors by measuring the co-occurrence frequency of papers by different authors. Figure 6 plots the co-citation network of the 100 most cited authors in each time slice (slice = 1) after the networks were simplified by using the pathfinder pruning algorithm. The nodes in the author co-citation network represent the authors, the size of the node is positively related to the number of citations, the links represent the co-citation relationship between authors, and the thickness of the line is positively related to the co-occurrence frequency of the authors. The author co-citation network used in this study included 518 nodes and 1841 links. The nodes are presented in the form of citation tree rings, which indicate the citation history. The colour of the tree ring represents the corresponding citation time, and the thickness of each tree ring is proportional to the number of citations in the corresponding time zone.

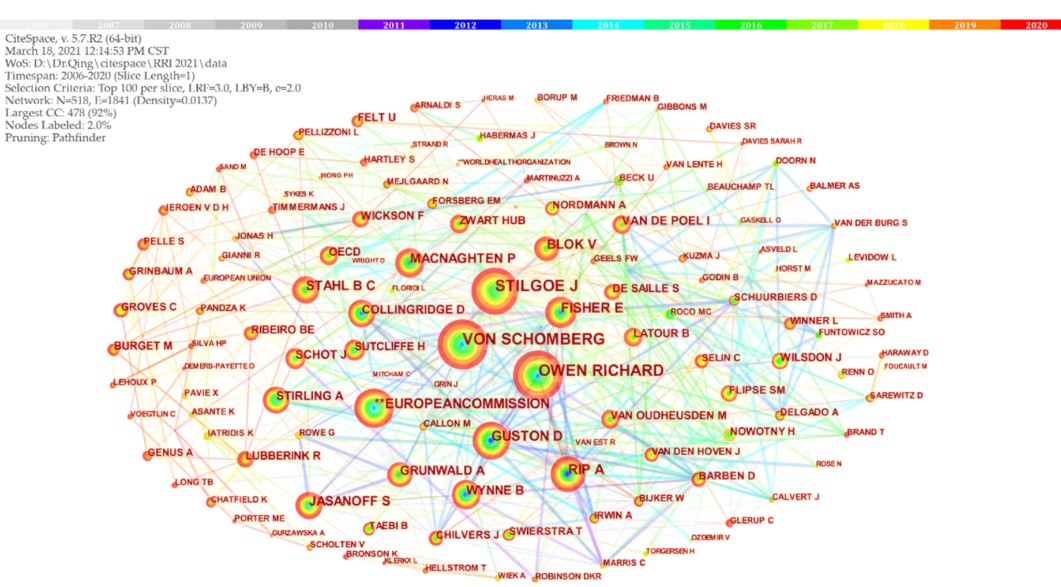

**Figure 6.** Author co-citation network for RRI research from 2006 to 2020.

The six largest nodes with a citation frequency of more than 100 in Figure 6 indicated that the leading scholars in the RRI research field are Jack Stilgoe, Richard Owen, René von Schomberg, Erik Fisher, and David Guston. The EC is the only institution among the highly cited authors, not surprising given that the term 'RRI' has been introduced into the research field in a top-down fashion by science policymakers and funding agencies mostly in the EC [45]. Owen's and Stilgoe's citation frequencies and number of co-occurrences top the list, mainly because the two participated in the formulation of the Framework for Responsible Innovation [2]. Fisher and Guston both work at the Center for Nanotechnology in Society at Arizona State University. Von Schomberg has been with the EC since 1998. Thus, RRI is receiving attention from both policy and academic circles.

## 5.2. Reference Co-Citation Analyses

### 5.2.1. Key Nodes in Reference Co-Citation Network

Reference co-citation analyses describe the connections between documents, which are an important tool for exploring the structure and evolution of a research field [46]. To more clearly visualise the time agglomeration of far-reaching documents in the RRI field, the time slice was set to 3. Figure 7 shows the reference co-citation network for RRI before 2020, which contained 358 nodes and 377 edges. The nodes in Figure 7 represent the references cited by the documents in the dataset, and the size of the nodes was positively correlated with the number of times the documents were cited. Similar to the author co-citation network, these nodes are depicted in the form of tree rings. The thicker the citation tree ring, the higher the number of citations in the corresponding time slice. The connection between nodes represents the co-citation relationship between them, and the time of the first co-citation is represented by the colour of the connection.

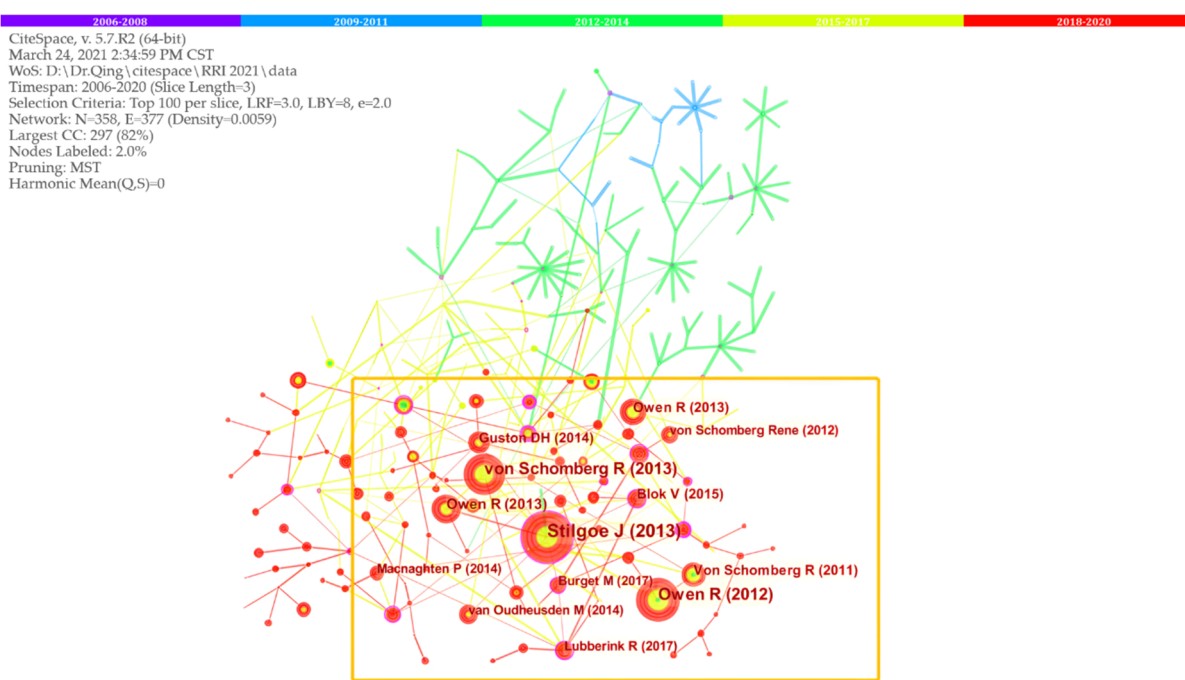

**Figure 7.** Reference co-citation network for RRI research from 2006 to 2020.

Before 2011, research on RRI was at an exploratory stage, paving the way for subsequent research. The nodes in Figure 7 represent landmark references (indicated by the box). Most of these documents were published in 2012–2014 and were cited frequently after 2015, laying the foundation for subsequent RRI research. This study selected three key nodes in the co-citation network for analysis, as shown in Table 1.

**Table 1.** The top three cited references for RRI during 2006–2020.

| Key Nodes | Content Category | Core View | Perspective |
|---|---|---|---|
| Jack Stilgoe et al. (2013) [2] | Definition | RI means taking care of the future through collective stewardship of science and innovation in the present | Foresight Process |
| | Dimension | Anticipation/Reflexivity/Inclusion/Responsiveness | |
| R. Owen et al. (2012) [47] | Feature | • Democratising the governance of the purpose of R&I <br> • Integrating and institutionalising reflective, anticipatory, and inclusive deliberative mechanisms in R&I processes <br> • Reframing responsibility for the R&I process and taking collective responsibility | Motivation Process Governance subject |
| von Schomberg (2013) [1] | Definition | RRI is a transparent, interactive process by which societal actors and innovators become mutually responsive to each other with a view to the (ethical) acceptability, sustainability, and societal desirability of the innovation process and its marketable products (so as to allow a proper embedding of scientific and technological advances in our society) | Process Product Results evaluation |
| | Mechanism | • Technology assessment and foresight <br> • Application of the precautionary principle <br> • Normative/ethical principles to design technology <br> • Innovation governance <br> • Stakeholder involvement and public engagement | |

Based on the citation frequency, three key-node documents from 2006 to 2020 were extracted, and the main ideas and research perspectives in these papers are summarised in Table 1. These articles focused on the discussion of the definition and characteristics of RRI. Stilgoe, et al. [2] defined RRI in a forward-looking perspective as 'taking care of the future through collective stewardship of science and innovation in the present', and proposed four dimensions of RRI, based on a process management perspective: anticipation, reflection, inclusion, and responsiveness (AIRR framework).

Owen, et al. [47] summarised three emergent features of RRI. Firstly, the purpose of R&I is 'for society'—to address societal challenges and achieve the 'right impact' on society, based on consultative democracy. Secondly, the process of R&I must be 'with society'—integrating reflective, anticipatory, and inclusive deliberative mechanisms to achieve responsive institutionalisation. Thirdly, the responsibility for R&I needs to be

'reframed'—reflecting the responsibilities of different roles (e.g., scientists, business partners and policymakers) in the innovation process, which should try to respond collectively.

Von Schomberg [1] defined RRI as 'a transparent, interactive process' based on a process perspective, and emphasised that RRI should focus on 'the acceptability, sustainability, and social expectations of the innovation process and its marketable products', based on a product and outcome perspective. He also listed four types of irresponsible innovation: (1) technology-driven, (2) ignoring basic ethical principles, (3) policy-pull, and (4) lack of precautionary measures and technological foresight.

These three influential RRI-related documents share similar elements in their conclusions, yet demonstrate heterogeneity in their dimensions and perspectives. In terms of dimensions, Stilgoe, et al. [2] and Owen, et al. [47] exhibited a consensus on four dimensions of RRI: anticipation, reflexivity (reflection), inclusion (inclusive deliberation), and responsiveness. Two of these dimensions, anticipation and inclusion, were also addressed in von Schomberg's definition [1], and his explicit reference to the dimension of 'sustainability' was also implicit in Jack Stilgoe et al.'s definition [2]. In terms of perspective, the process perspective was addressed in all three articles, suggesting that the governance of the R&I process is an inescapable focus when examining the definition and connotations of RRI. The studies by Stilgoe, et al. [2] and von Schomberg [1] both reflected a forward-looking perspective. In addition, Owen, et al. [47] focused on two important aspects of RRI, motivation and governing subject, while von Schomberg [1] defined RRI primarily from a product and outcome perspective.

### 5.2.2. Reference Citation Burst Analysis

Citation burst analysis is conducted to detect landmark articles with a high growth rate of citations. If an article's citation frequency increases rapidly within a certain period, it is marked as having a strong citation burst. Articles with a strong citation burst represent an academic turning point in a certain field; they are indicators that the scientific community is paying close attention to a particular concept. Figure 8 shows the top 21 references with the strongest citation bursts during the period between 2006 and 2020 (sorted by the beginning year of the burst). Detailed information on these 21 publications (sorted by strengths of the burst) is provided in Appendix A Table A1.

According to the burst strength analysis, Stirling's research [56] had the strongest citation burst, lasting for four years from 2013 to 2016 and demonstrating a burst strength of 7.92. This research explored the relationship between power and the normative, instrumental, and substantive imperatives in social appraisal, and, on this basis, concluded that technology governance needs to place greater emphasis on open participatory deliberation. Stirling's comprehensive examination of the contrastive implications of the normative, instrumental, and substantive imperatives has had a profound impact on the RRI field. It provided for further understanding of the normative, instrumental, and substantive motivations of social appraisal based on Fiorino's research [66] and has had far-reaching implications [67].

Fisher, et al.'s work [48] had the second strongest citation burst, with its strength calculated as 7.09 and a span from 2010 to 2014. This article proposed a 'midstream modulation' of RRI and emphasised 'reflexive awareness' in the research and development (R&D) process—that is, the timely adjustment of R&D activities through reflection on realities and values. Midstream modulation, public upstream engagement, and downstream governance (such as regulations and markets) together constitute the three-stage governance system of authorisation (upstream), implementation (midstream), and adoption (downstream) in the R&D process.

Some articles showed burst status for several years in a row, although the burst strength was not very robust. Four articles had the (same) longest burst durations, lasting six years, and all of them were highly regarded and cited by the academic community within a short period of time after publication, so they had weaker citation bursts [68]. The common feature of these articles was that they explored the theoretical framework [25,50]

and practical approach [42,53] of RRI to some extent (even if the concept of RRI was not widely used at the time of publication, the connotations of the ideas wehomologous to RRI).

## Top 21 References with the Strongest Citation Bursts

| References | Year | Strength | Begin | End | 2006 – 2020 |
|---|---|---|---|---|---|
| Barben D, 2007, HANDBOOK OF SCIENCE AND TECHNOLOGY STUDIES, V0, P979 | 2007 | 4.86 | **2009** | 2014 | |
| Fisher E, 2006, B SCI TECHNOL SOC, V26, P485, | 2006 | 7.09 | **2010** | 2014 | |
| Fisher Erik, 2007, NANOETHICS, V1, P155 | 2007 | 5.98 | **2010** | 2013 | |
| Schuurbiers D, 2009, EMBO REP, V10, P424, | 2009 | 4.84 | **2010** | 2015 | |
| van der Burg S, 2009, SCI ENG ETHICS, V15, P97, | 2009 | 3.81 | **2010** | 2014 | |
| Owen R, 2010, RISK ANAL, V30, P1699, | 2010 | 5.45 | **2011** | 2016 | |
| Macnaghten P, 2011, NATURE, V479, P293, | 2011 | 4.13 | **2012** | 2016 | |
| Robinson DKR, 2009, TECHNOL FORECAST SOC, V76, P1222, | 2009 | 3.09 | **2012** | 2017 | |
| Matthew Kearnes, 2009, JENSEITS REGULIERUNG, V0, P97 | 2009 | 2.85 | **2012** | 2016 | |
| Wynne B, 2006, COMMUNITY GENET, V9, P211, | 2006 | 2.78 | **2012** | 2014 | |
| Stirling A, 2008, SCI TECHNOL HUM VAL, V33, P262, | 2008 | 7.92 | **2013** | 2016 | |
| Rip A, 2009, EMBO REP, V10, P666, | 2009 | 3.25 | **2013** | 2016 | |
| Stegmaier P, 2009, EMBO REP, V10, P114, | 2009 | 2.89 | **2013** | 2014 | |
| Jasanoff S, 2009, MINERVA, V47, P119, | 2009 | 4.19 | **2015** | 2017 | |
| EuropeanCommission, 2013, OPT STRENGTH RESP RE, V0, P0 | 2013 | 3.77 | **2015** | 2017 | |
| Owen R, 2013, RESPONSIBLE INNOVATI, V0, P0 | 2013 | 2.99 | **2015** | 2016 | |
| Strand R, 2015, INDICATORS PROMOTING, V0, P0 | 2015 | 3.22 | **2016** | 2017 | |
| Grunwald A, 2011, ENTERPRISE WORK INNO, V7, P9 | 2011 | 3.19 | **2016** | 2017 | |
| Burget M, 2017, SCI ENG ETHICS, V23, P1, | 2017 | 5.82 | **2018** | 2020 | |
| STAHL BC, 2017, SUSTAINABILITY-BASEL, V9, P0, | 2017 | 3.78 | **2018** | 2020 | |
| Ribeiro BE, 2017, SCI ENG ETHICS, V23, P81, | 2017 | 3.05 | **2018** | 2020 | |

**Figure 8.** Top 21 references with the strongest citation bursts over the period between 2006 and 2020 [15,25,42,48–65].

In addition, the citations of three articles published in recent years are continuing to burst, and they are expected to have a significant impact on future academic trends. The first article, which outlines Stahl et al.'s Maturity Model [64], provides a reference guide for the practice of RRI in corporate settings and serves as a bridge between theory and reality. RRI theory has been largely developed by the policy community, grown by the academic community, and will inevitably be deeply integrated into the R&I process in the future. The other two papers are literature reviews. Burget, et al. [15] provide an overview of the conceptual dimensions of RRI, intending to build a broader empirical basis. More comprehensively, Ribeiro, et al. [65] summarise the key dimensions of RRI—motivations, theoretical conceptualisations, and translations into practice—and distil the different conceptual frameworks associated with them.

### 5.2.3. Identification of Hotspots through Citation Clustering

Figure 9 shows the clustering results for the RRI-related literature from 2006 to 2020, with the network-related parameters noted in the upper-left corner of the image. The clusters were extracted by using indexing terms and the cluster labels were defined by the log-likelihood ratio (LLR) algorithm. The cluster labels represent the main research topics in the field of RRI, and their accuracy and validity can be judged to some extent by the parameters. On the one hand, the 'Modularity Q' of 0.6595 was significant and

relatively high, indicating that the network was reasonably divided into several loosely coupled modules. On the other hand, the 'weighted mean silhouette' of 0.8164 (greater than 0.7) implied that the homogeneity of each cluster is high [35]. Specifically, the top 11 clusters all had high silhouette scores, indicating that the clustering results were plausible (see Appendix A Table A2 for more details).

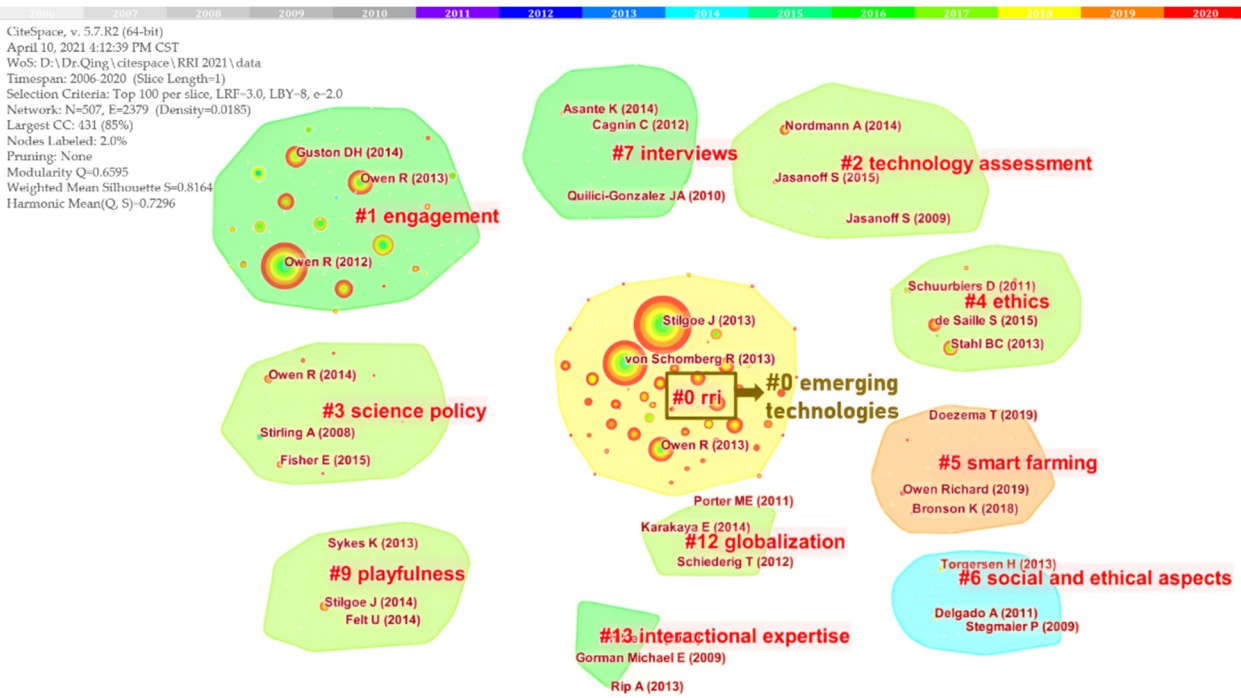

**Figure 9.** Clustering map for cited references related to RRI research from 2006 to 2020 [1,2,31–33,47,56,58,59,61,69–90].

In terms of cluster size, Cluster 0 was the largest and was defined as 'RRI' (LLR = 12.23), the subject and search term of this paper, which was sensible but did not provide much guidance. Cluster 0 was therefore identified as comprising the second highest LLR subject term, 'emerging technologies' (LLR = 8.45). Thus, the first five clusters were 'emerging technologies', 'engagement', 'technology assessment', 'science policy', and 'ethics'. Subsequent studies analysed these in detail.

- Cluster 0: Emerging Technologies

The largest cluster, Cluster 0, was labelled Emerging Technologies, and signalled the inextricable interaction between emerging technologies and RRI. On the one hand, the development of emerging technologies laid the foundation for RRI. The 'dual use' of emerging technologies [2] has led to an increasingly widespread debate on the topic of responsibility for innovation [11,91]. Governance models that rely solely on state regulation have unavoidable drawbacks, such as the time lags associated with policies [92]. In response, an open and decentralised governance model, that is more adapted to emerging technologies, has emerged. On the other hand, emerging technologies are the governance object of RRI. To avoid the irreversible effects of emerging technologies [93], RRI considers their ethical acceptability and extant social needs, and offers a governance concept that is more focused on a process perspective, such as advocating upstream public engagement [94,95] and interdisciplinary cooperation [96,97].

As technology changes and new technologies emerge, application of RRI is being advocated for an increasing number of industries and scenarios. A few typical examples are given here.

The first example is the use of RRI to address the developmental challenges facing synthetic biology. Synthetic biology is an emerging field that lies between biology and engineering; it advocates the construction of unnatural biological modules and systems for various scenarios, such as energy, materials, and medical applications, and has been described as a continuation of early genetic engineering. However, synthetic biology faces concerns, similar to those associated with 'genetically modified organisms' (GMOs), mainly in regard to safety issues (e.g., synthetic pathogens leaking into the environment, creating biological weapons), ethical issues (e.g., weakening the distinction between organisms and machines), and socio-economic issues (e.g., distribution of benefits and risks) [98]. The ELSA approach introduced earlier has been widely applied for governance purposes in synthetic biology, but has been criticised for focusing too much on outcomes and neglecting practical processes [99]. In contrast, the RRI concept, with its emphasis on public engagement and process management, has been more gradually applied to synthetic biology. Specific strategies include improving risk assessment; training researchers in RRI behaviour and ethics; coordinating interdisciplinary collaborations among academia, government, and industry; and improving knowledge-sharing platforms [100].

A second example is the use of the RRI approach to mitigate conflicts within the semi-autonomous vehicles industry, such as the attribution of liability for traffic accidents, data privacy issues, the protection of insurers' interests, and so on. The RRI approach is used throughout the negotiation process to accompany and guide insurers, vehicle manufacturers, policymakers, and other stakeholders [101]. The heterogeneity of RRI practices in different industries and contexts enriches its methodology and is expected to provide guidance for wider practice in the future.

- Cluster 1: Engagement

The discussion of 'engagement' is another hot topic in the RRI field. Practical experience with public engagement (PE) in R&I, in forms such as consensus conferences and citizens' juries, has a long history predating the RRI movement [2]. Likewise, inclusion is a key dimension in RRI definitions and has been widely agreed upon by scholars [47]. As a core aspect of the RRI framework, PE is required to be broad and inclusive [15]. To include the widest range of diverse people in the interaction process of R&I, the EU-funded initiative Public Engagement Innovations for Horizon 2020 (PE2020) was proposed. The findings of PE2020 show that the trend of increasing involvement of organised stakeholder groups (e.g., environmental and industrial organisations) in R&I continues, while the 'fourth sector', composed of hybrid experts, randomly selected participants, 'life world experts', and 'field experts', is becoming more prominent [102].

Further, according to the RRI framework's principles of anticipation and responsiveness, PE should occur in the upstream stages of policy development and implementation [77]. In a general sense, upstream engagement is considered a powerful tool for bridging the gap between science and society. An editorial in Nature asserted that upstream engagement, if properly managed, would not end any field of research [103]. However, later scholars, following the development of teams using upstream engagement, found that upstream researchers were concerned about this model of handing over control to 'outsiders', mainly in terms of limiting early research inspiration and the loss of advantages from delaying the development process [104].

To make upstream engagement more effective, a PE model that adopts both foresight and a deliberative approach has been implemented and has been proven to improve the quality of the decision-making process [34]. Based on the same aspirations, Fisher, et al. [48] advocated the integration of midstream modulation and upstream engagement, thereby creating a form of collaborative research between natural scientists and engineers, on the one hand, and social scientists and humanist scholars, on the other hand [12]. Referring to the motivation or impact of PE, three categories can be distinguished: normative (e.g., to achieve democracy and equity), instrumental (e.g., to achieve pre-committed policy goals), and substantive (e.g., to generate tangible products, to mobilise social resources) [47,56,102].

Stakeholders are a pivotal group in the broad concept of PE [102]. Stakeholder engagement is considered a key approach to RRI, as well as an important practice to meet the 'inclusion' and 'reflexivity' expectations in the RRI framework [2,61,105]. Similar to the case for PE, stakeholder engagement advocates the inclusion of members of society beyond the immediate beneficiaries in the technological development framework, thus ensuring that the R&I process is open, transparent, and democratic. Some projects have explored different forms of stakeholder participation [106–108].

However, every issue has two sides. In concrete practice, stakeholder engagement has also encountered barriers. Van der Meij, et al. [108] argue that the lack of participants' expertise, combined with a lack of communication experience among R&I practitioners, can make the engagement process challenging. Van de Poel, et al. [109] summarised four types of barriers to stakeholder engagement in RRI projects: lack of resources, cognitive limitations, confidentiality issues, and lack of trust. To overcome these barriers, they proposed targeted solutions, such as organising more stakeholder engagement activities, broadening assessments, focusing on peer competition issues, and performing early engagement activities. The importance of stakeholder engagement in RRI has been thoroughly proven over the years. Identifying barriers to stakeholder engagement in different RRI projects and designing more flexible and efficient engagement mechanisms to better integrate participants in the R&I process are current and future research priorities.

- Cluster 2: Technology Assessment

The roots of both ELSA and RRI can be primarily traced back to TA, an endeavour that was institutionalised in 1972 with the establishment of the Office of Technology Assessment in the United States [110]. In its early stages, traditional TA relied on quantitative data and expert analysis to predict and avoid the potentially harmful effects of emerging technologies, a strategy also known as parliamentary TA [111]. An intentional effort to overcome reliance on quantitative and top-down expert models, participatory TA emerged in Europe and emphasised a greater focus on broad stakeholder engagement, which inspired the emergence of the CTA approach. The CTA approach shares common ground with RRI in its advocacy of the co-evolutionary theory of science and society and its promotion of the social embedding of innovation.

CTA evolved from TA, the main thrust of which was to develop strategies and tools to feed the results of TA back into the actual construction of technologies, thereby broadening the design of new technologies and changing the design of old ones [27]. CTA places great emphasis on the design of the technology being evaluated, and its core philosophy asserts the need to include a large number of participants who have differing perspectives, including social scientists, citizens, NGOs, research funders, policymakers, innovative firms, and other stakeholders; which is a stance that meshes well with RRI. Thus, CTA is, in a sense, an approach to implement the RRI framework [112]. It urges researchers to proactively identify possible deviations of supply and demand [113] and potential barriers in the innovation process through stakeholder engagement [75], and advocates for the development of technologies in socially desirable directions. CTA has been widely used in emerging technological fields, such as nanotechnology [79], medicine [112], and battery technology [114], due to the open and dynamic nature of the process and its highly inclusive definition of stakeholders [115].

- Cluster 3: Science Policy

RRI, which has emerged as the most popular science policy in Europe and beyond in recent years [116], challenged the traditional neutralist science policy [74], by embodying the notion of democratised [47], decentralised [117], sustainable [118] science and technology (S&T) governance. With a focus on the governance of R&I's purpose and intent, and advocating for the shaping or monitoring of technology through broad PE, RRI represents strong opposition to the trend towards depoliticization [119]. Before the RRI movement, initiatives such as STS, TA, and ELSA (ELSI) were policies and programmes geared towards socio-technical integration. This category of science policy, which sought to integrate social

aspects into the R&D process producing emerging technologies, marked the expansion of the object of S&T governance from risk and impact to science and technology itself [120]. Changes in the socio-technical integration of science policy at the macro level have an effect on research solicitations and research performance at the meso-micro level [14,67].

The trend of RRI entering mainstream consciousness in the research world has become increasingly evident since its inclusion in Horizon 2020 as a cross-cutting theme. After Horizon 2020 came to an end, Horizon Europe, which succeeded it, did not identify RRI as a stand-alone theme, but reflected the key dimensions and basic ideas of RRI at an in-depth level. Linden Farrer, working for the Directorate-General for Research and Innovation of the EC, argued that Horizon 2020 positioned RRI as mainstream thinking and that Horizon Europe integrated RRI as an expected project practice. In Horizon Europe, the RRI framework exists in a higher work plan than it did in Horizon 2020 and is an ongoing concern that does not end [121]. In the open science section, for example, co-design, co-creation, and co-assessment are proposed as PE activities that are expected to have a profound impact on the institutional governance of the participating beneficiaries [122].

- Cluster 4: Ethics

Technological innovation has ethical challenges in a universal sense, which are caused by extensive social relations and complex environmental changes [123]. Unlike traditional innovation ethics research, which focuses more on the professional ethics of innovators, RRI covers a wider range of stakeholders and places more emphasis on embedding ethical management in the R&I process, thereby incorporating both ethics and society into innovation activities [101,124].

In basic research projects, ethical management is designed as an embedded module in the project to promote RRI. The EU-funded Human Brain Project (HBP), for example, has a dedicated ethics sub-project SP12, a research ecosystem of ethics experts, ethics reporters, and an ethics registry. All of its work is carried out in close cooperation with researchers from the various sub-projects of the HBP [125,126]. This organisational structure ensures that ethics management is professional, deeply participatory, and open, and avoids the awkward situation in which ethics management is, in practice, a separate sub-project.

In business organisations, the other mainstay of R&I, —the ethical objectives of innovators tend to be market-oriented, such as customer satisfaction, organisational reputation, and legal compliance. Such concerns are far removed from the academic ambitions promoted by RRI. Although in the short-term, addressing noble ethical challenges is far removed from more utilitarian organisational goals, developing a responsible view of corporate social relations and a culture of ethical awareness is key to organisational longevity [127]. RRI has led to a greater awareness of responsibility and ethics, but the sense of responsible ownership among researchers has yet to be strengthened. Researchers should be conscious of the negative impacts expected during the innovation process, and not just look to regulators and public involvement for monitoring and solutions.

*5.3. Journal Co-Citation Analysis*

The journal co-citation network presents the structure and distribution of the knowledge base by measuring the frequency of co-occurrence of journals published by the reference. Figure 10 plots the co-citation network of the 100 most cited journals in each time slice (slice = 1) after simplification with the pathfinder pruning algorithm. Similar to the previous analyses, the nodes in the journal co-citation network represented journals, and edges represented co-citation relationships between journals. Figure 10 shows that the knowledge base of the RRI study was mostly concentrated in journals, including Research Policy, JRI, Science and Public Policy, and Science and Engineering Ethics. Notably, Responsible Innovation is a pivotal book in the knowledge base, first published in 2013. It has been an important research foundation for the RRI field since then.

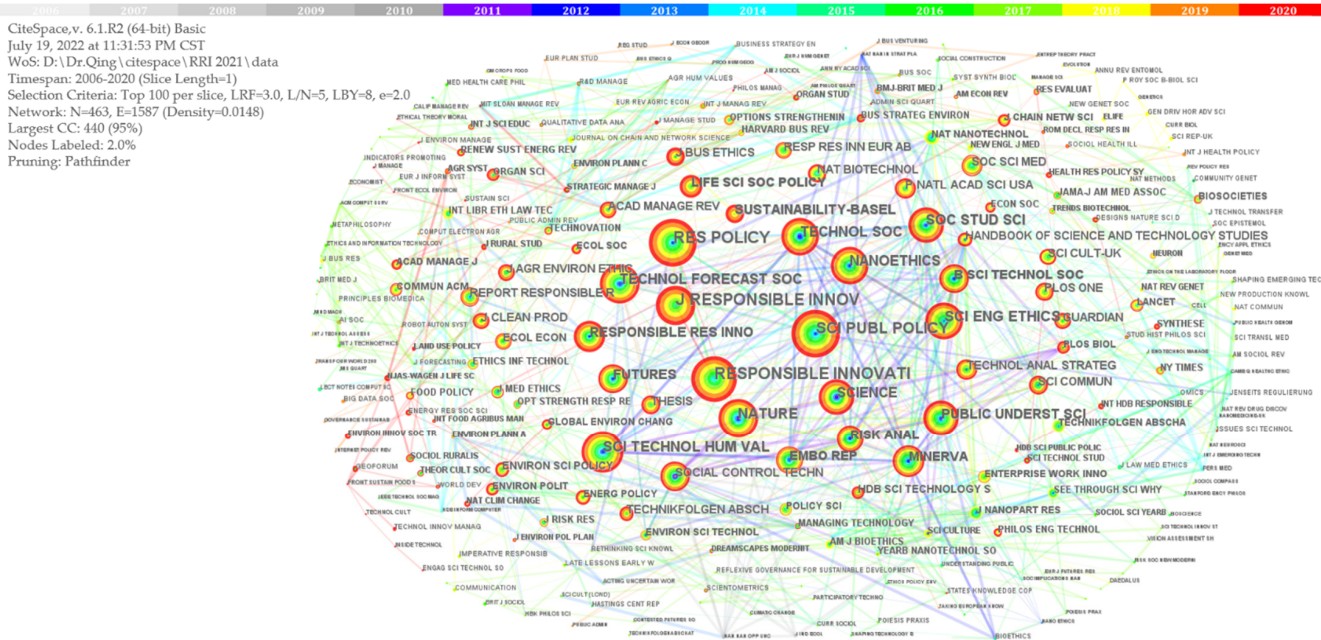

**Figure 10.** The journal co-citation network for RRI research from 2006 to 2020.

Specifically, among the top five influential publications, JRI and Responsible Innovation are specialised publications focusing on RRI research, while the other journals include public policy and ethics research. Figure 11 shows the annual citation trends for these five publications, with an overall trend of rapid growth in these citations after 2014 being apparent. In addition, Research Policy consistently received high citations, even surpassing those of the two RRI-specific publications, JRI and Responsible Innovation, after 2018. There are two main reasons for this. Firstly, Research Policy has been in existence for a long time and has a high impact in the field of policy research. Secondly, several highly cited, seminal works in RRI research have been published in Research Policy, such as 'Developing a Framework for Responsible Innovation'.

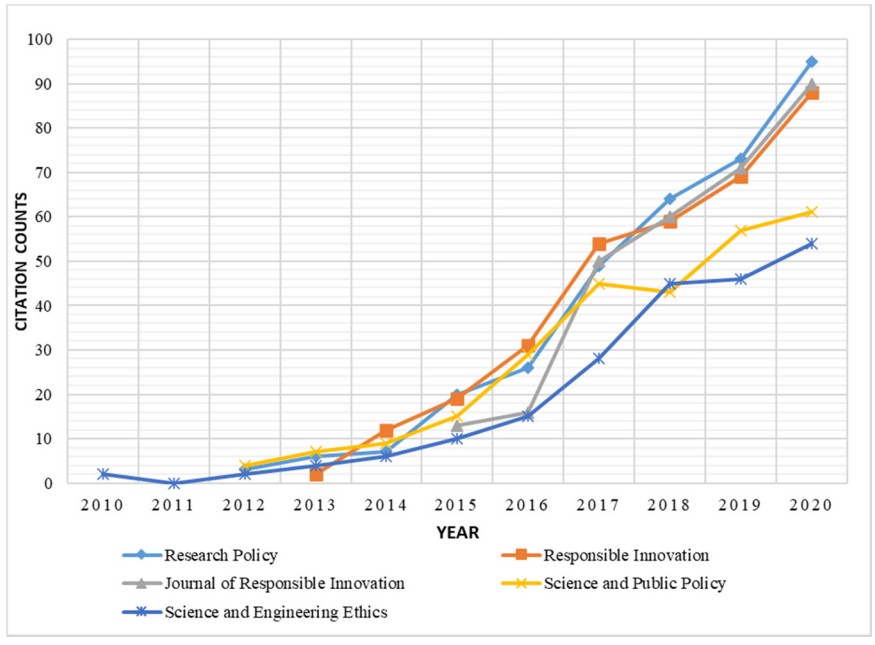

**Figure 11.** Citation counts for the top five publications.

### 6. Co-Authorship Analysis about RRI Research

The institution and country co-authorship networks describe the academic links between different institutions and countries. Figure 12 shows the top 100 institutions with the most published RRI research in each time slice, including 129 active nodes and 149 edges. As can be seen in the figure, Delft University of Technology, Arizona State University, De Montfort University, and Oxford University are the top research institutions in terms of the number of publications.

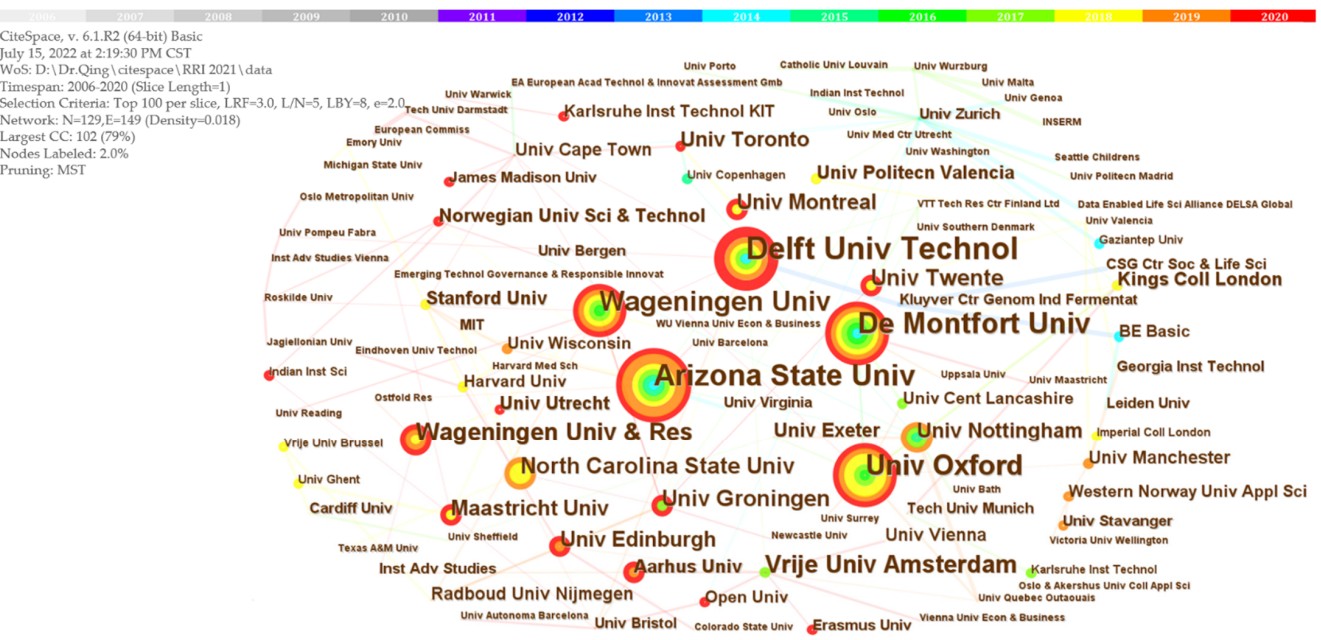

**Figure 12.** The institution co-authorship network for RRI research from 2006 to 2020.

These results also indicated that RRI research is highly valued in both European countries and the United States, which was further validated by the national co-authorship network shown in Figure 13 and more visually represented in Figure 14. Beyond this, RRI research has not yet become widespread in other continents and is dominated by developed countries and a few major developing countries. The Netherlands, the United Kingdom, and Germany are the countries with the highest academic contributions to RRI research in Europe. Despite this, these countries show a low centrality in the network and researchers in these countries do not collaborate very frequently. In contrast, the United States has relatively extensive collaboration with other countries in RRI research. Overall, the majority of RRI research is in Europe and North America, with a tendency to spread to other regions.

The results suggested that RRI is more widely discussed in technologically advanced places. On the one hand, this finding occurs because RRI is relevant to some ethically controversial frontier technologies, such as nanotechnology and genetic modification. On the other hand, these countries and regions have a long history of research on the ethics of science and technology and have a deep and solid academic foundation.

In recent years, RRI has also gradually attracted attention in developing countries, though researchers there have shown some discomfort with this framework. For example, Gao, et al. [128] pointed out the lack of a dialogue mechanism that would support RRI practices in China, based on research. More importantly, the priority task for developing countries is to try to reduce the technology gap relative to developed countries, rather than to take social responsibility for their research. RRI practice, therefore, needs to be located and engaged within local contexts, cultures and practices [129].

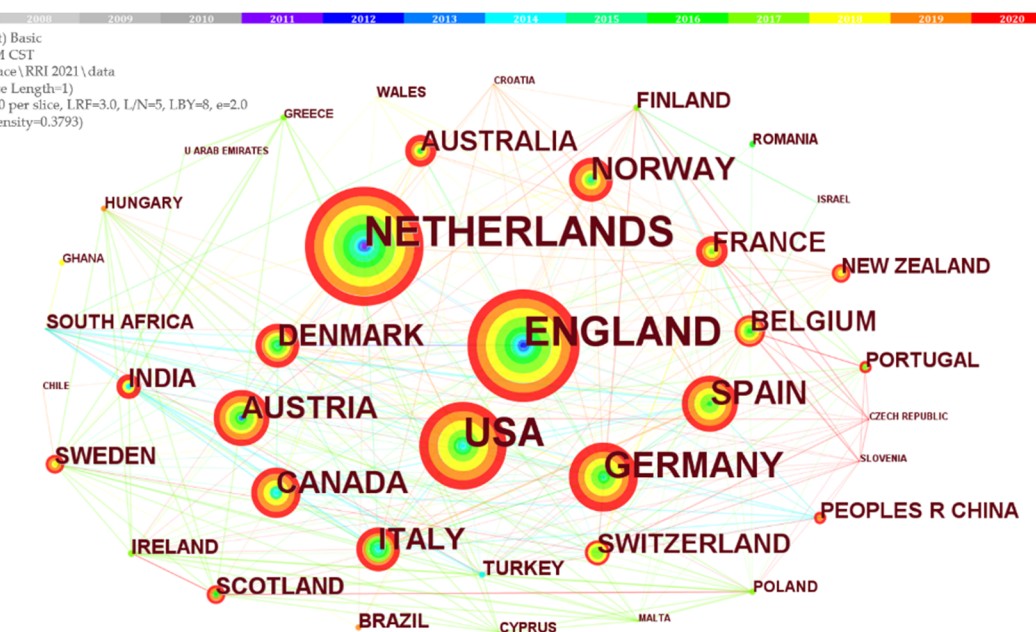

**Figure 13.** The country co-authorship network for RRI research from 2006 to 2020.

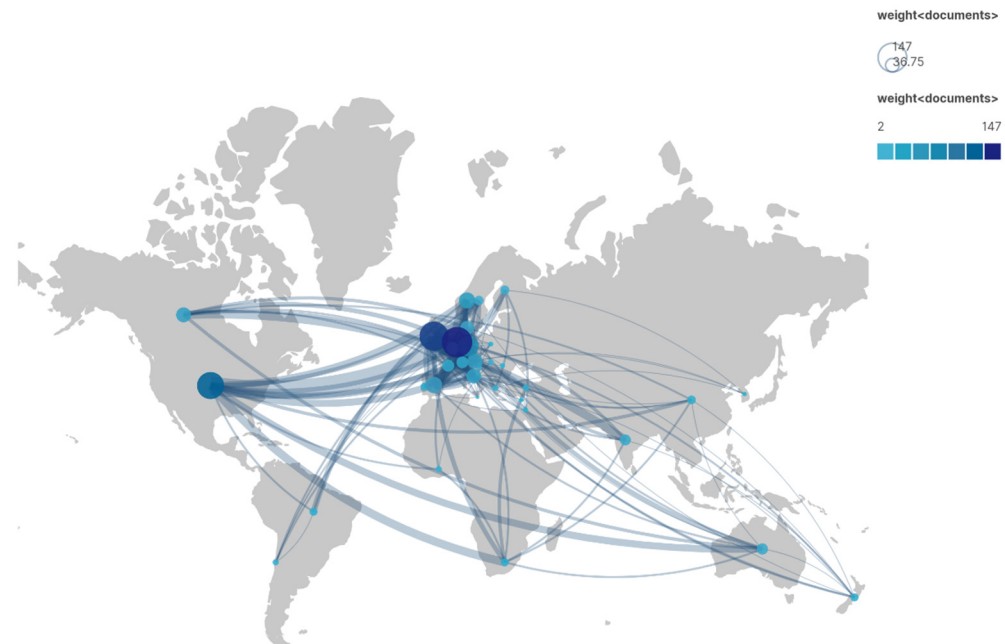

**Figure 14.** The country co-authorship network of RRI research from 2006 to 2020 plotted on the world map.

## 7. Phased Keywords Analysis of RRI Research

Keywords are a distillation and summary of the core content of an article and are an important tool to provide an entire perspective on the field of study. In this research, keyword co-occurrence and Timezone visualisation were used to track the thematic evolution of RRI domains [130]. Due to the small number of articles published before 2010. and the absence of the co-occurrence keyword in those pre-2010 articles, this paper selected other keywords between 2010 and 2020. The time slice was set to four years (the last phase was three years, due to data limitations), thereby distinguishing three development phases. The visualisation results are shown in Figure 15 and the three phases of the RRI thematic

evolution analysis are summarised in Table 2. Due to their high statistical frequency, the search terms 'responsible innovation' and 'responsible research and innovation' appeared too large in the figure to allow for the display of the other keywords, so these two terms are hidden. The other keywords are arranged by frequency in the time slice.

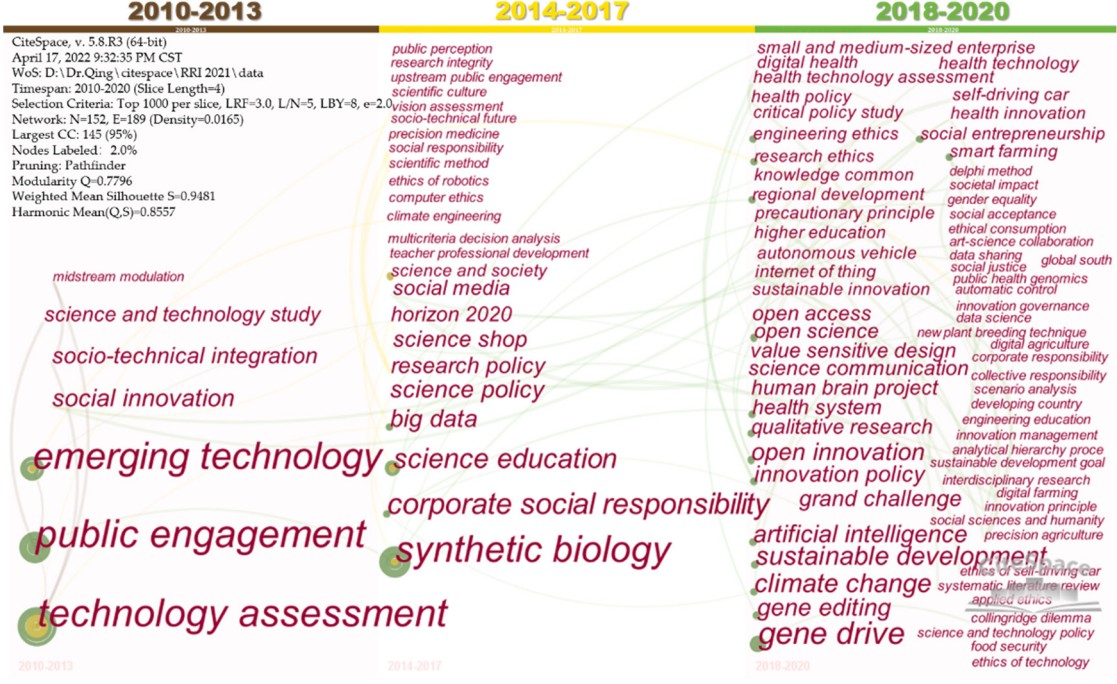

**Figure 15.** Keyword co-occurrence in the Timezone visualisation of RRI research from 2010 to 2020.

**Table 2.** The evolutionary trajectory of the RRI field: a summary of the characteristics of the three phases.

| | Phase I: Theory Development Phase 2010–2013 | Phase II: Initial Application Phase 2014–2017 | Phase III: Application Expansion Phase 2018–2021 |
|---|---|---|---|
| **Topics and Related Keywords** | Innovation and Society <br>• Social innovation <br>• Socio-technical integration <br>• Science and technology study <br>• Emerging technology | Initial Application to Specific Areas <br>• Synthetic biology <br>• Big data <br>• Climate engineering <br>• Ethics of robotics <br>• Computer ethics <br>• Precision medicine | Wide Range of Applications in Specific Areas <br>• Artificial intelligence <br>• Data sharing <br>• Data science <br>• Internet of Things <br>• Automatic control <br>• Human Brain Project <br>• Smart farming <br>• Digital agriculture <br>• Precision agriculture <br>• Digital farming <br>• Self-driving car <br>• Autonomous vehicle <br>• Gene drive <br>• Gene editing <br>• Health innovation <br>• Health policy <br>• Health technology assessment <br>• Health system |

Table 2. *Cont.*

| | Phase I:<br>Theory Development Phase<br>2010–2013 | Phase II:<br>Initial Application Phase<br>2014–2017 | Phase III:<br>Application Expansion<br>Phase<br>2018–2021 |
| --- | --- | --- | --- |
| | RRI Core Methodology<br>• Public engagement<br>• Midstream modulation<br>• Technology assessment | Development of Tools and Methods<br>• Multicriteria decision analysis<br>• Science shop<br>• Vision assessment<br>• Upstream public engagement<br>• Scientific method<br><br>Extension to the Private Sector<br>• Corporate social responsibility<br>• Sustainability transition<br><br>Initial Focus if the EU's RRI Values<br>• Science education<br>• Teacher professional development | Enrichment of Tools and Methods<br>• Delphi method<br>• Analytical hierarchy process<br>• Value-sensitive design<br>• Interdisciplinary research<br>• Sociotechnical integration<br>• Art–science collaboration<br>• Qualitative research<br><br>Reaching out to Developing Countries<br>• Developing country<br>• Global South<br><br>Comprehensive Focus of the EU's RRI Values<br>• Gender equality<br>• Higher education<br>• Open access<br>• Open science<br>• Open innovation<br>• Social justice<br>• Research ethics<br>• Applied ethics<br>• Ethics of technology<br>• Sustainable innovation<br>• Sustainable development<br>• Sustainable development goal<br>• Innovation governance<br>• Regional development |
| **Main Research directions** | • Background,<br>• Connotations,<br>• Frameworks,<br>• Methods of RRI | • Practice in specific areas<br>• Tools and methods<br>• New implementation subjects | • Practice in specific areas<br>• Tools and methods<br>• New countries/culture context<br>• More value dimensions |

• Phase I: Theory Development Phase (2010–2013)

The years 2010–2013 saw the initial development of RRI, with a relatively small number of papers published, and therefore a smaller distribution of keywords. These keywords all revolved around theoretical research on RRI, and could be broadly divided into two categories. Firstly, the keywords 'social innovation', 'socio-technical integration', and 'science and technology study' pointed to the integration of society and technology, which was the core concept of RRI and inherited the goal of 'technology assessment' [56]. Secondly, the core tools and methods of RRI were another focus of this phase. 'Public engagement' has a long history of addressing the inclusion dimension of RRI; with regards

to the reflection dimension, the 'midstream modulation' framework, as an interdisciplinary approach, aims to facilitate scientists' reflections on social ethics [48,73].

As noted earlier, the seminal literature in the field of RRI was published in Phase I, and that discourse on the background, connotations, frameworks, and methods of RRI has had a profound impact on subsequent research. Therefore, the first phase is labelled as the 'theory development phase'. The colour of the dots and lines of the keywords in Phase I show that these keywords have never faded from the research horizon of RRI and are closely linked to other keywords, indicating that the theoretical connotations and core methods have always been the focus of research in the field of RRI and will be enriched and improved in the future as RRI practice evolves and scenarios expand.

- Phase II: Initial Application Phase (2014–2017)

The year 2014 was important in the evolution of RRI, with a significant increase in the rate of publication growth occurring since then, which was inseparable from such contributing factors as the creation of JRI and the launch of Horizon 2020. Based on the bibliometric results, the study summarises several important features of Phase II.

Firstly, RRI has seen initial applications in technical areas, such as synthetic biology [100,131], big data [132], climate engineering [133,134], and computing [135].

Secondly, some methods and tools have been developed or applied to RRI practice. Multicriteria decision analysis was used to address issues in the RRI governance process [136] and Science Shop was applied to the RRI field as a tool to facilitate public engagement [137].

Thirdly, the private sector, outside of publicly funded projects, is beginning to focus on RRI. The implementation of RRI has helped to realise the aspirations of corporate social responsibility (CSR) [138].

Fourthly, the EC brings a richer perspective to RRI but is less researched [65]. Science education has attracted academic attention in Phase II and aims to transparently engage all societal players in R&I by promoting science education and facilitating access to scientific knowledge.

In conclusion, RRI practices were initially developed in Phase II, mainly in terms of the enrichment of application areas, tools and methods, and subjects of implementation. Thus, Phase II is labelled as the 'initial application phase' of RRI.

- Phase III: Application Expansion Phase (2018–2021)

After 2018, efforts aimed at expanding the scope of RRI became more abundant. In Figure 15, it is obvious that the number of keywords detected in Phase III was significantly higher than in Phases I and II. Due to space limitations, it is difficult to analyse all the keywords one by one. However, after categorising and summarising them, this study selected several categories of research themes that did not appear in the previous two phases for analysis and summarised the new features of Phase III.

Firstly, in Phase III, RRI became widely used in more technical fields. These research areas include artificial intelligence [139,140], the HBP [125,126], smart farming [141,142], autonomous vehicles [143,144], data science [145], health innovation [146,147], and gene drives [148–150], among others.

Secondly, more specific methods were developed or applied to RRI practice. Umbrello [151] added heuristic tools to value-sensitive design (VSD) methods to eliminate the negative effects of moral intuitions. The Delphi method was used as a forecasting process framework to analyse the potential positive and negative impacts of virtual fencing (VF), aiming to apply the forward-looking principles of RRI to this technique [152]. Ko, et al. [153] used the analytical hierarchy process (AHP) approach to explore the benefits and costs of implementing RRI in firms, arguing that RRI can be replicated in profit-first firms.

Thirdly, the influence of RRI began to radiate to developing countries beyond its birthplace, taking into account the different cultural contexts and political environments in these countries relative to the West [154]. For example, Setiawan [155] explored the impact of higher power distance and collectivism on RRI dimensions based on the case

of CO2 utilisation in Indonesia, arguing that this factor is informative for RRI practices in developing countries with similar cultural contexts.

Fourthly, the RRI values promoted by the EC were given more attention and studied in Phase III, as reflected by the emergence of the following keywords: gender equality; higher education; open access, open science, open innovation; social justice; research ethics, ethics of technology; sustainable innovation, sustainable development, sustainable development goal, climate change; innovation governance.

In summary, the RRI field made overall progress in Phase III in terms of application areas, tools, context, research areas, and other dimensions. Phase III is therefore labelled as the 'application expansion phase'.

## 8. Evolutionary Trajectory and Future Research Directions

Based on the phased keywords analysis, combined with the results of the previous series of analyses, this study summarised the evolutionary trajectory of the RRI field along the logical line 'theory–practice–subject–context' and identified potential themes for future research.

### 8.1. Theory

The dimensions of RRI are undergoing a process of continuous enrichment. RRI is the current representative policy for the concept of socio-technical integration and has developed classic connotative dimensions, such as the AIRR framework, based on the critical inheritance of past policies and programmes, such as STS, TA, and ELSA. With the introduction and implementation of the EU funding programme, Horizon 2020, RRI completed its mainstreaming process. Horizon Europe, the current EU funding programme, has integrated RRI into its practice, and there is reason to believe that the connotations and dimensions of RRI will further expand and evolve in the future in response to technological and social developments and changes in social values.

### 8.2. Practice

The technical fields in which RRI is implemented are constantly broadening. The analysis of the distribution of publications and disciplines revealed that RRI is typically interdisciplinary, simultaneously covering the ethics of technology, public policy, and specific technical areas. The fields of synthetic biology, artificial intelligence, big data, autonomous vehicles, smart farming, and climate engineering are among the technical areas in which RRI has most recently been implemented. This study considers the practice of RRI in emerging or sustainability-related technologies as a key area for future research.

Accordingly, more methods and tools are now being applied, albeit sometimes tentatively, to RRI practices. There is currently a trade-off between flexibility and standardisation in specific RRI approaches, with some industries expecting a flexible RRI approach to suit their domain, while standardised processes are more applicable in other industries. RRI guidance offering more adequate information and examples remains lacking, however [156]. This study, therefore, believes that industry-specific design concepts, that can be applied to practice, will be the focus of future RRI exploration.

### 8.3. Subject

RRI is gradually moving from government-funded projects to the private sector, such as in for-profit businesses. The application of RRI to the R&D process by companies is a way of achieving CSR. Embedded ethical management approaches have been used to promote RRI, especially in government-funded basic research projects. For example, in the HBP, the ethics management system works closely with the project researchers, rather than having ethics management as a separate sub-project. This can serve as a valuable reference point for other organisations' RRI implementation efforts [125].

RRI emphasises collective responsibility [1,47]. As business practices are an important subject of R&I outside the publicly funded sector, and a critical component of social

engagement, there has been considerable experience with, and research on, RRI practices in the business sector [109,153]. However, the ethical goals of business tend to be more market-oriented [127], and CSR may serve as a bridge between business organisations and RRI [157]. This suggests that future researchers should combine and build on the findings and experiences related to CSR to further standardise the dimensions and methods of RRI practice in the context of profitability of business organisations and to strengthen the sense of responsibility of R&D personnel.

*8.4. Context*

RRI is gradually spreading to developing countries beyond its birthplace. Nevertheless, differences between countries in terms of cultural background, political environment, and level of development are likely to affect the acceptance and difficulty of practising RRI in specific contexts. For example, existing research suggests that characteristics such as high-power distance and collectivist culture have an impact on the RRI dimension [155]. Moreover, developing countries, such as China, for example, are more concerned with closing the technological gap with developed countries than taking responsibility for innovation. In addition, the lack of a development platform and communication mechanism for RRI is an important issue that needs to be addressed in developing countries [128]. This study believes that future RRI practice mechanisms can be improved by exploring ways to make flexible adaptations, based on local cultural and economic developments.

## 9. Discussion

RRI is an innovative governance concept that has received widespread attention in political and academic circles. Considering the diversity of the RRI literature, a bibliometric approach was adopted to visualise the current state of research, hotspots and trajectory in this field, and to predict future directions. The findings are well placed to answer the three research questions posed in the article. Firstly, this study depicted the current research status of RRI from multiple perspectives (RQ1). On the one hand, the number distribution, journal distribution, and disciplinary distribution visually depicted the profile of the RRI field. On the other hand, co-authorship analyses portrayed the collaborative network of the RRI field, at both institutional and national levels. Secondly, the paper explored the hotspots of RRI through various co-citation analyses (RQ2), the most important of which was the reference co-citation analyses, including key literature analysis, citation burst analysis and citation clustering analysis. Finally, combined with the above analyses, this study made an attempt to predict the future direction of RRI (RQ3), based on the combing of evolutionary trajectory of RRI. The keyword-based phased thematic evolution analysis is a powerful tool for analysing the evolutionary trajectory of RRI. In summary, this study has theoretical and practical implications for the development of the RRI field.

Firstly, this study sorted out and compared the more widely accepted definitions of RRI to provide some context for the current lack of consensus on the definition of RRI in academia. Through the reference co-citation analysis, the majority of the seminal literature in this field was found to have been published in 2012–2014. Combined with the findings of the keyword-based thematic evolution analysis, this period roughly coincides with Phase I (theory development phase), which focused on the background, connotations, framework, and methodology of RRI. By analysing several key nodes in the reference co-citation analysis, this study found that the early highly cited literature focused on exploring the connotations and conceptual framework of RRI and was highly recognised by later researchers. The relatively small number of definitions of RRI produced during Phase I originated not only from academia, but also from policymakers. The AIRR framework, published in the academic journal Research Policy, has also been widely cited by subsequent researchers. The definition proposed by Von Schomberg (A staff member of the EC) included elements such as inclusiveness, participation, anticipation, societal desirability, and ethical acceptability, which are closely related to the EU's policy values. What both the AIRR framework and the EC definition have in common is that they conceptualise

RRI as incorporating the dimensions of inclusion and anticipation. Thus, there is a high level of consensus that public engagement and risk identification are key aspects of RRI research. In terms of the current state of RRI research, uniformity and clarity in the definitions and dimensions are generally lacking, but the findings suggest that the most cited definition, and, hence, the major point of consensus, is the AIRR framework proposed by Stilgoe, et al. [2].

Secondly, the results of both the reference co-citation cluster and keywords analyses pointed to the importance of PE in the RRI field. PE is an indispensable practice of RRI and the strategic focus of the CTA method, which aims to promote openness and democracy within the R&I process. From the perspective of participants, the inclusion dimension in the RRI framework requires PE to be broad and diverse. The findings of the PE2020 show that there is an increasing trend towards diversity of participants. In terms of motivations for engagement, three categories can be distinguished: normative (e.g., to achieve democracy and equity), instrumental (e.g., to achieve pre-committed policy goals), and substantive (e.g., to generate tangible products, to mobilise social resources). In terms of the engagement phase, upstream engagement is accepted by academics as a powerful tool for innovation or technology governance. However, research on upstream engagement has revealed certain disadvantages of RRI practices, such as limiting inspiration and delaying the R&I process. The upstream–midstream–downstream three-stage governance system proposed by Fisher, et al. [48] integrates science and society more comprehensively at a theoretical level. In terms of barriers to engagement, van de Poel, et al. [109] summarised four categories of barriers to stakeholder engagement from the RRI project—lack of resources, cognitive limitations, confidentiality issues, and lack of trust—and proposed targeted measures to address them. As a core practice of RRI, the importance of PE has been widely agreed upon by the policy and academic communities, and future research should pay attention to designing more flexible and improved mechanisms that can accommodate different situations.

Finally, based on the analysis of the landscape mapping and evolutionary trends related to RRI, this study proposes the following practical implications. Firstly, the disciplinary distribution, subject burst analysis, and keyword analysis point to the use of RRI in industries, such as big data, artificial intelligence, synthetic biology, and smart agriculture, as current or potential research hotspots. RRI as a governance principle is bound to encounter trade-offs between standardisation and flexibility in different scenarios or fields. This study proposes that researchers and organisations focus on the practice of RRI in emerging technologies or sustainable development-related fields and design more industry-specific practice processes. Secondly, the results of the reference co-citation clustering and keywords analyses suggest that there is a growing interest in RRI from commercial organisations, but the government-funded DNA of RRI does not match the profit-seeking nature of such organisations. This study, therefore, suggests that it is necessary to focus on business-led RRI practices and that RRI incentives take into account the profitability needs of businesses. Thirdly, the results of the country co-authorship and keywords analysis suggest that the influence of RRI has spread to developing countries, a trend that is well worth promoting and encouraging. However, existing RRI concepts and frameworks are not always fully applicable in these new environments. This study suggests focusing on the practice of RRI in emerging economies and adapting the dimensions and practices of RRI to the local cultural and political contexts.

## 10. Limitations

This study has some limitations. Firstly, the data sample was limited. Although the sample came from the largest database of publications available (Web of Science), papers written in languages other than English and unpublished papers were not included. Future RRI studies are encouraged to cover more comprehensive sample sets to expand and complement the research.

Secondly, the existing bibliometric methods have limitations. On the one hand, the current bibliometric tools cannot accurately identify the contributions of authors other than

the first author. On the other hand, self-citations and negative citations cannot be excluded, and the results are biased to a certain extent [158]. This study suggests that future studies undertake more detailed and accurate landscape studies of RRI with the help of more advanced bibliometric methods.

**Author Contributions:** Conceptualization, J.L. (Jiqing Liu) and J.L. (Jiayu Li); software, X.L.; formal analysis, J.L. (Jiqing Liu); writing—original draft preparation, J.L. (Jiqing Liu); writing—review and editing, G.Z. and J.L. (Jiayu Li); supervision, J.L. (Jiqing Liu); funding acquisition, G.Z. All authors have read and agreed to the published version of the manuscript.

**Funding:** This research was funded by National Social Science Fund of China, grant number 18ZDA044. The APC was funded by Gui Zhang.

**Institutional Review Board Statement:** Not applicable.

**Informed Consent Statement:** Not applicable.

**Data Availability Statement:** The data presented in this study are available in article.

**Conflicts of Interest:** The authors declare no conflict of interest.

## Abbreviations

| | |
|---|---|
| AIRR framework | anticipation, reflection, inclusion, and responsiveness |
| CSR | corporate social responsibility |
| CTA | constructive technology assessment |
| EC | the European Commission |
| ELSA | ethical, legal, and social aspects |
| ELSI | ethical, legal, and social implications |
| EPSRC | Engineering and Physical Sciences Research Council |
| EU | the European Union |
| HBP | Human Brain Project |
| JRI | Journal of Responsible Innovation |
| LLR | log-likelihood ratio algorithm |
| PE | public engagement |
| PE2020 | Public Engagement Innovations for Horizon 2020 |
| R&I | research and innovation |
| RI | Responsible Research and Innovation |
| RRI | Responsible Innovation |
| S&T | science and technology |
| STS | science and technology studies |
| TA | technology assessment |

## Appendix A

**Table A1.** Details of top 21 references with the strongest citation bursts.

| References | Topic | Strength | Duration |
|---|---|---|---|
| Stirling, A., 2008, *Sci. Technol. Hum. Values, 33*, 262 [52] | 'Opening up' and 'closing down': Power, participation, and pluralism in the social appraisal of technology | 7.92 | 4 |
| Fisher, E., 2006, *Bull. Sci. Technol. Soc., 26*, 485 [48] | Midstream modulation of technology: Governance from within | 7.09 | 5 |
| Fisher, E., 2007, *NanoEthics, 1*, 155 | Ethnographic invention: Probing the capacity of laboratory decisions | 5.98 | 4 |
| Burget, M., 2017, *Sci. Eng. Ethics, 23*, 1 [15] | Definitions and conceptual dimensions of responsible research and innovation: A literature review | 5.82 | 3 |

**Table A1.** *Cont.*

| References | Topic | Strength | Duration |
|---|---|---|---|
| Owen, R., 2010, *Risk Anal., 30*, 1699 [42] | Responsible innovation: A pilot study with the UK Engineering and Physical Sciences Research Council | 5.45 | 6 |
| Barben, D., 2008, *The Handbook of Science and Technology Studies, 0*, 979 [25] | Anticipatory governance of nano-technology: Foresight, engagement, and integration | 4.86 | 6 |
| Schuurbiers, D., 2009, *Embo Rep., 10*, 424 [50] | Lab-scale intervention | 4.84 | 6 |
| Jasanoff, S., 2009, *Minerva, 47*, 119 [59] | Containing the atom: Sociotechnical imaginaries and nuclear power in the United States and South Korea | 4.19 | 3 |
| Macnaghten, P., 2011, *Nature, 479*, 293 [52] | Good governance for geoengineering | 4.13 | 5 |
| van der Burg, S., 2009, *Sci. Eng. Ethics, 15*, 97 [51] | Imagining the future of photoacoustic mammography | 3.81 | 5 |
| Stahl, B.C., 2017, *Sustainability-Basel, 9*, 1036 [64] | The Responsible Research and Innovation (RRI) maturity model: Linking theory and practice | 3.78 | 3 |
| European Commission, 2013 [60] | Options for strengthening responsible research and innovation | 3.77 | 3 |
| Rip, A., 2009, *Embo Rep., 10*, 666 [57] | Futures of ELSA | 3.25 | 4 |
| Strand, R., 2015, *Indicators for promoting and monitoring responsible research and innovation* [62] | Expert group on policy indicators for responsible innovation: Indicators for promoting and monitoring responsible research and innovation report from the Expert Group on Policy Indicators | 3.22 | 2 |
| Grunwald, A., 2011, Enterprise and Work Innovation Studies, *31*, 10–31. [63] | Responsible innovation: Bringing together technology assessment, applied ethics, and STS research | 3.19 | 2 |
| Robinson, D.K.R., 2009, *Technol. Forecast. Soc. Change, 76*, 1222 [53] | Co-evolutionary scenarios: An application to prospecting futures of the responsible development of nanotechnology | 3.09 | 6 |
| Ribeiro, B.E., 2017, *Sci. Eng. Ethics, 23*, 81 [65] | A mobilising concept? Unpacking academic representations of responsible research and innovation | 3.05 | 3 |
| Owen, R., 2013 [31] | Responsible innovation: Managing the responsible emergence of science and innovation in society. | 2.99 | 2 |
| Stegmaier, P., 2009, *Embo Rep., 10*, 114 [58] | The rock 'n' roll of knowledge coproduction | 2.89 | 2 |
| Kearnes, M., 2009, Jenseits von Regulierung: Zum politischen Umgang mit der Nanotechnologie, 97 [54] | The emerging governance landscape of nanotechnology | 2.85 | 5 |
| Wynne, B., 2006, *Public Health Genomics, 9*, 211 [55] | Public engagement as a means of restoring public trust in science: Hitting the notes, but missing the music? | 2.78 | 3 |

**Table A2.** The cited references cluster details in Figure 9.

| Cluster ID | Cluster Label | Cluster Size | Silhouette | Mean (Year) |
| --- | --- | --- | --- | --- |
| 0 | Emerging technologies | 90 | 0.688 | 2015 |
| 1 | Engagement | 86 | 0.714 | 2011 |
| 2 | Technology assessment | 53 | 0.887 | 2013 |
| 3 | Science policy | 51 | 0.863 | 2013 |
| 4 | Ethics | 41 | 0.813 | 2012 |
| 5 | Smart farming | 28 | 0.966 | 2017 |
| 6 | Social and ethical aspects | 27 | 0.918 | 2009 |
| 7 | Interviews | 24 | 0.935 | 2012 |
| 9 | Playfulness | 20 | 0.948 | 2012 |
| 12 | Globalisation | 6 | 0.995 | 2011 |
| 13 | Interactional expertise | 5 | 0.989 | 2011 |

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
