# Peer review of "Discovering the Landscape and Evolution of Responsible Research and Innovation (RRI): Science Mapping Based on Bibliometric Analysis"

_sustainability, doi:10.3390/su14148944_

Round 1

Reviewer 1 Report

Congratulation on your paper.

The title is too long.

The aim is clear.

The references are very representative

The Introduction/background is clear and evaluates well the state of art.

The research question clearly outlined and relevant for the subject

The methods are appropriately described and applied. 

Figures 4 - 12  have too small resolution. Are unclear. 

Table 1 is very clear and representative for the paper.

The article comes with new and surprising ideas in the field

I appreciate very much the cluster analysis

The result is not a repetitive text and emphasize what is statistically significant.

I appreciate that authors added a practically meaningful result

The results discussed from multiple angles and placed into context without being overinterpreted.

The conclusions are based on the aims of the study.

The limitations of the study are important, but brings opportunities to inform future research

Overall, the article is consistent within itself.

Author Response

Dear Reviewer:

Please see the attached file for a detailed response. 

Thank you.

Reviewer 2 Report

The study reviews and assesses the development of RRI research via a bibliometric analysis of 702 RRI-focused papers and 26,471 secondary references published in the 15 Web of Science Core Collection database in 2006-2020.

The aim of the study is well defined, making it relevant.

The abstract is informative enough.

The introduction, Literature review and research design are well structured

The results are clearly presented. The arguments in the discussion section are compelling.

Conclusions are clearly structured and well presented.

Congrats! 

Author Response

Dear Reviewer:

Thank you very much for your recognition of our article and we will continue to work towards a better paper.

Best wishes.

Reviewer 3 Report

In my opinion, it is a very good work that is close to being finished. My suggestion is mainly a formal one. The figures are difficult to see at this size and are not informative enough. I suggest that they be uploaded as an additional file at full size.

My other suggestion is to answer the research questions (Q1, Q2, Q3) at the beginning of the article in the same structure at the end. It helps the clarity of the article if the structure is clear. This is not a comment on the content, just a suggestion on how to present the results. 

Author Response

(The authors gave the same response as above.)

Reviewer 4 Report

The paper "Discovering the Landscape and Evolution of Responsible Research and Innovation (RRI): Science Mapping Based on Bibliometric Analysis" is recommended for minor revision.

The paper has been well prepared, and has good flow.

The paper has good organisation and structure.

The paper has been well referenced and introduced.

The paper has good discussion of results and good analysis of the mapped networks.

The only concern is that it requires minor proofreading and some English language editing.

In technical paper writing, the use of personal or collective pronouns like "we" and "our" are not allowed in academic writing as they were used alot in this paper, but it should be revised please. It is seen mostly in all from Section 1 - Section 10 and in abstract.

In Abstract, line 16 "First, we provide a broad..." should be rewritten, such as "Firstly, the paper was prepared by providing a broad..."

Also, in the abstract, the words First, Second, Third used in lines 16,18,19 should be Firstly, Secondly, Thirdly.... 

Improve abit on the conclusions made and also discuss how the geographical location influences the evolution of RI and RRI.

To ensure the tool used is validated - CiteSpace, add some references for CiteSpace. Visit the website, and see some and include them, such as the following:

Chen, C. (2004) Searching for intellectual turning points: Progressive Knowledge Domain Visualization. Proc. Nat. Acad. 945 Sci., 101(Suppl.), 5303-5310. 946

Chen, C. (2006) CiteSpace II: Detecting and visualizing emerging trends and transient patterns in scientific literature. 947 JASIST, 57(3), 359-377. 948

Chen, C. (2010) System and method for automatically generating systematic reviews of a scientific field. US Patent 949 US20110295903A1. 950

Chen, C. (2016) CiteSpace: A Practical Guide for Mapping Scientific Literature. Nova Science Publishers. 951

Chen, C. (2017) Science mapping: A systematic review of the literature. JDIS, 2(2), 1-40. https://doi.org/10.1515/jdis-2017-952 0006 953

Chen, C. (2020) A Glimpse of the First Eight Months of the COVID-19 Literature on Microsoft Academic Graph. Front. Res. 954 Metr. Anal. 5:607286. 955

Chen, C. et al. (2010) The structure and dynamics of co-citation clusters: A multiple-perspective co-citation analysis. JASIST, 956 61(7), 1386-1409. 957

Chen, C., Song, M. (2019) Visualizing a field of research: A methodology of systematic scientometric reviews. PLoS ONE, 958 14(10): e0223994. https://doi.org/10.1371/journal.pone.0223994

Author Response

(The authors gave the same response as above.)
